# Difficult Airway Assessment Based on Multi-View Metric Learning

**DOI:** 10.3390/bioengineering11070703

**Published:** 2024-07-11

**Authors:** Jinze Wu, Yuan Yao, Guangchao Zhang, Xiaofan Li, Bo Peng

**Affiliations:** 1School of Computing and Artificial Intelligence, Southwest Jiaotong University, Chengdu 611756, China; jzwu@my.swjtu.edu.cn (J.W.); xfl@my.swjtu.edu.cn (X.L.); 2General Practice Medical Center, West China Hospital, Sichuan University, Chengdu 610044, China; yaoyuan@wchscu.cn; 3Department of Anesthesiology, West China Hospital, Sichuan University, Chengdu 610044, China; zhangguangchao@wchscu.cn

**Keywords:** deep learning, difficult airway assessment, multi-view metric learning, complementarity and consistency of information

## Abstract

The preoperative assessment of difficult airways is of great significance in the practice of anesthesia intubation. In recent years, although a large number of difficult airway recognition algorithms have been investigated, defects such as low recognition accuracy and poor recognition reliability still exist. In this paper, we propose a Dual-Path Multi-View Fusion Network (DMF-Net) based on multi-view metric learning, which aims to predict difficult airways through multi-view facial images of patients. DMF-Net adopts a dual-path structure to extract features by grouping the frontal and lateral images of the patients. Meanwhile, a Multi-Scale Feature Fusion Module and a Hybrid Co-Attention Module are designed to improve the feature representation ability of the model. Consistency loss and complementarity loss are utilized fully for the complementarity and consistency of information between multi-view data. Combined with Focal Loss, information bias is effectively avoided. Experimental validation illustrates the effectiveness of the proposed method, with the accuracy, specificity, sensitivity, and *F*_1_ score reaching 77.92%, 75.62%, 82.50%, and 71.35%, respectively. Compared with methods such as clinical bedside screening tests and existing artificial intelligence-based methods, our method is more accurate and reliable and can provide a reliable auxiliary tool for clinical healthcare personnel to effectively improve the accuracy and reliability of preoperative difficult airway assessments. The proposed network can help to identify and assess the risk of difficult airways in patients before surgery and reduce the incidence of postoperative complications.

## 1. Introduction

With the rapid development of medical technology, difficult airway management has become a significant challenge in the fields of anesthesiology and emergency medicine. A difficult airway refers to situations where healthcare professionals with appropriate airway management skills encounter difficulties when performing tracheal intubation or mask ventilation on patients. These difficulties may be caused by the patient’s anatomical structure and pathological characteristics or unexpected situations during the operation. Some patients may even have difficulty during extubation [1]. The incidence of difficult intubation ranges from 1.9% to 10%, while the incidence of difficult mask ventilation is between 1.4% and 5.0% [2,3,4,5,6,7]. These high incidences pose a serious threat to patients’ safety. An unexpectedly difficult airway is one of the main factors that lead to patients being unable to intubate and unable to ventilate. In such cases, the patient’s oxygenation status will drop sharply, which may lead to severe hypoxemia, leading to hypoxic brain damage or even death [8,9]. To reduce the risks of difficult airways, the preoperative assessment of difficult airways is crucial. Through preoperative assessment, doctors can timely identify the most difficult airway situations, allowing them to take necessary preventive measures before surgery or other operations. This can reduce the occurrence of unexpectedly difficult airways and the associated risks and challenges after anesthesia [10,11].

Clinically, bedside screening tests are commonly used for preoperative assessments, which include the measurements of mouth opening, thyromental distance, head and neck mobility, the upper and lower lip bite test, and the Mallampati score [12,13], etc. However, these preoperative assessment methods mainly rely on the subjective judgment of doctors, which is significantly influenced by their personal experience and skill level, leading to a large degree of subjectivity and randomness. In addition, these methods can only provide partial information and are unable to comprehensively and accurately assess the patient’s airway status. Due to these clear defects, the diagnostic accuracy and reliability are relatively low [3,14].

In recent years, with the rapid development of artificial intelligence technology, especially the widespread applications of deep learning in the medical field, new insights have been found to solve such problems. The facial image-based difficult airway assessment is a prospective research direction pursued by many scholars. Relevant studies have shown that patients with difficult airways exhibit subtle differences in facial features compared to those with non-difficult airways. Therefore, modeling and analyzing these differences through artificial intelligence methods can more objectively and accurately identify the risk factors for difficult airways. These methods can enhance the accuracy and reliability of preoperative assessments, providing better safety for patients. Some scholars have already analyzed facial images based on artificial intelligence methods to assess difficult airways. Tatsuya Hayasaka et al. [15] used a modified VGG16 model as a classification model to train and classify the facial images of patients from 16 different angles. The optimal model was trained based on images of patients in a supine position with their mouths closed. They utilized class activation mapping to visualize the model, intuitively showing the model’s focus during the identification process, thus verifying the model’s effectiveness. Tavolara et al. [16] cropped patient facial images according to regions like the eyes, nose, and mouth. They used ensemble learning to train individual regions’ images separately in convolutional neural network models and identified difficult airways through a majority voting mechanism. Wang et al. [17] proposed a fast, non-invasive, cost-effective, and highly accurate deep learning method. They utilized the Mixmatch semi-supervised learning algorithm to train a semi-supervised deep learning model on facial image data of patients in five poses. This model can achieve comparable results to fully supervised models with only a small amount of labeled data, and the cost of sample labeling is lower. García-García et al. [18] proposed an algorithm that utilizes airway morphological information to reliably predict intubation risks. They used deep convolutional neural networks to extract key anatomical landmarks from patients’ preoperative photos and obtained 3D coordinates of these landmarks through 3D modeling. Based on these coordinates, a series of respiratory tract morphological features were calculated and finally, machine learning methods were used to classify these morphological features. This method significantly outperforms expert judgments and existing state-of-the-art methods in predicting intubation risks, especially in reducing false negatives.

The facial image-based difficult airway assessment relies on information from multiple views of the patient. The complementarity and consistency of information between multi-view data are the key to difficult airway assessments. However, the above or most existing methods are based on machine learning, single-view-based deep learning, or simple multi-view-based deep learning, which fail to fully explore and utilize the complementarity and consistency of information between multi-view data. The accuracy and reliability of assessments remain low, and difficult airway assessments are still a challenging task.

In this paper, we aim to explore deep-learning methods to improve the accuracy and reliability of difficult airway assessments. A novel deep learning model called the Dual-Path Multi-View Fusion Network is proposed, which employs a dual-path parallel structure to extract multi-scale features from frontal and profile facial images of patients and performs feature fusion. Moreover, consistency and complementarity loss functions are used to constrain the extracted image features. We incorporated a Hybrid Co-Attention Module to make the model focus on regions of interest, such as the face and neck while ignoring background noise. Different from the existing difficult airway recognition algorithms, we designed consistency loss and complementarity loss based on the multi-view metric learning method, explored the information consistency and complementarity between the multi-view data, made full use of the rich information contained in the multi-view data, and improved the accuracy and reliability of the difficult airway assessment.

Our research motivation is to improve the accuracy and reliability of difficult airway assessments by introducing this novel method, thereby providing medical staff with more reliable auxiliary tools, reducing the occurrence of postoperative accidents, and ensuring patient safety. Specifically, the main contributions are as follows:A new network model is specifically designed for difficult airway assessment tasks.The proposed method combines multi-view learning and metric learning, fully utilizing the complementarity and consistency of information between multi-view data to improve the model’s accuracy and reliability.Clinical data of multi-view facial images are collected to validate the methods. Experimental evaluation demonstrates the efficiency of the proposed method.

The remainder of this paper is organized as follows: Section 2 details the dataset and proposed methodology in this research. Section 3 presents the experimental results. Section 4 discusses the effectiveness and reliability of our approach. Finally, Section 5 concludes the research.

## 2. Materials and Methods

This section first introduces in detail the dataset used in our study, including the collection standards and methods of the datasets. Then, the image preprocessing steps are described, including center cropping and data augmentation. Four data augmentation methods used in this study are listed. Finally, the structure and sum of the proposed model are described in detail.

### 2.1. Dataset

Starting from June 2023, we conducted an observational study on 320 patients who underwent general anesthesia surgery at the hospital. With written consent from each patient, we collected facial images of each patient, including the following five views: the frontal closed-mouth base position, frontal open-mouth base position, frontal tongue-protruding base position, lateral closed-mouth base position, and lateral closed-mouth chin-lift position. Figure 1 shows some image data collected from each patient’s five different views. In these images, the periocular area of the patients was mosaiced to protect the privacy of the patients.

We selected patients for participation in this research based on four criteria: (1) patients were older than 18 years old; (2) patients had not undergone surgery to change their facial appearance or neck mobility; (3) patients had no central nervous system disease or psychiatric disease; and (4) patients were intubated without the use of other instruments during general anesthesia. During the process of tracheal intubation by the anesthesiologist after the induction of general anesthesia, we recorded the patients’ laryngoscope images, which were graded and recorded by anesthesiologists with more than three years of experience according to the Cormack–Lehane classification [19,20]. The definition of the Cormack–Lehane classification indicates the patient’s glottis during tracheal intubation, which includes five grades. Grade I indicates that the entire vocal cord is visible, Grade II indicates that only part of the vocal cord is visible, Grade III indicates that the epiglottis is visible, but the vocal cord is not, and Grade IV indicates that the epiglottis is not visible. Patients graded as I and II were considered non-difficult airway patients, while patients graded as III and IV were considered difficult airway patients. Based on these criteria, a total of 225 patients were included in this research, including 151 males and 74 females, with an average age of 52.4 years. Among them, 200 patients presented with normal airways, accounting for 88.9%, and 25 patients presented with difficult airways, accounting for 11.1%. All the images collected constituted the image dataset used in this research.

### 2.2. Data Preprocessing

Due to the limited amount of data, poor data quality, and imbalance between positive and negative samples in the dataset used in this research, we implemented a series of data preprocessing methods, as shown in Figure 2, to address these issues to a certain extent, improve the training effect of the model, and enhance the generalization ability of the model.

#### 2.2.1. Image Central Cropping

There is a lot of background noise in the images in the dataset. This noise not only reduces the prediction ability of the model but also increases the computational burden and training time of the model. In order to reduce the interference caused by background noise, we adopted a common preprocessing technique called central cropping. Central cropping is a simple but effective method that selectively crops the image to remove areas unrelated to the main task, i.e., background noise, allowing the model to focus more on learning the features of the target area. Specifically, for all views, the cropping range was set to include the patient’s forehead at the top, the bottom of the neck, and the edges of the face on the left and right. Such a cropping range was designed to retain the areas that were most relevant to difficult airway prediction—the patient’s face and neck—and remove most of the irrelevant background noise. The images are then resized to 224 × 224 pixels to fit the input of the model.

#### 2.2.2. Data Augmentation

The dataset contains 25 patients with difficult airways and 200 patients without difficult airways, resulting in a positive-to-negative sample ratio of 1:8. To address the imbalance between positive and negative samples, we adopted an offline data augmentation strategy. Specifically, for negative samples (non-difficult airways), the number of samples was increased to four times the original, reaching 800 samples. For positive samples (difficult airways), the number was increased to 16 times the original, reaching 400 samples. The resulting positive-to-negative sample ratio after data augmentation was 1:2. Our data augmentation methods include random horizontal flipping, random changes in brightness, contrast, and saturation, random rotations within the range of [−15°, 15°], and random erasing. Random horizontal flipping can change the direction of the image, increasing data diversity. Random changes in brightness, contrast, and saturation can simulate images under different lighting conditions, making the model more adaptive to changes in illumination. Random rotations can simulate images captured from different angles, enhancing the richness of the data. Random erasing actively eliminates some information in the image by randomly selecting a rectangular region and replacing its pixel values with random values. The above methods can effectively expand the dataset, increase the diversity of samples, enable the model to better learn the characteristics of the data, improve the generalization ability and robustness of the model, and reduce the risk of overfitting.

### 2.3. Model

In this section, we introduce the proposed Dual-Path Multi-View Fusion Network (DMF-Net), which adopts a specific dual-path structure to effectively extract multi-scale image features from frontal and lateral facial views. Firstly, we define the representation of a multi-view dataset as $x=\left\{X^{\left(1\right)},X^{\left(2\right)},X^{\left(3\right)},X^{\left(4\right)},X^{\left(5\right)}\right\}$, and the label of the multi-view data is represented as $y=\left\{y_{1},y_{2},\ldots ,y_{N}\right\}$, where $X^{\left(v\right)}\in R^{d_{v}\times N}$ represents the sample set of the $v^{th}$ view, $d_{v}$ represents the dimension of the single view sample, and N represents the number of samples. Based on front and side views, we divided the dataset into $x_{1}=\left\{X^{\left(1\right)},X^{\left(2\right)},X^{\left(3\right)}\right\}$ and $x_{2}=\left\{X^{\left(4\right)},X^{\left(5\right)}\right\}$, where $x_{1}$ is the set of three front views and $x_{2}$ is the set of two side views. The network fuses image features from multiple views and utilizes multi-view learning and metric learning to effectively utilize the complementarity and consistency of information between multi-view data, thereby improving the predictive ability and robustness of the model. Figure 3 shows the overall structure of the Dual-Path Multi-View Fusion Network, and Algorithm 1 shows the algorithmic flow of the model.
**Algorithm 1:** Algorithm of difficult airway classification with DMF-Net**Input:** multi-view data $\chi =\left\{X^{\left(1\right)},X^{\left(2\right)},X^{\left(3\right)},X^{\left(4\right)},X^{\left(5\right)}\right\};$ the label of data is $y=\left\{y_{1},y_{2},\ldots ,y_{N}\right\}$.
**Initialize:** randomly initialize model parameters $\theta_{FEM},\theta_{MFFM},\theta_{HCAM},\;\theta_{CPP},\theta_{CSP},\theta_{CLS}$.
**While** not converged, **perform the following**:
1:Concatenate three front views as $x^{1}$ and two side views as $x^{2}$ on the channel dimension;2:Obtain feature vectors $\left\{f_{1}^{gap},f_{2}^{gap}\right\}$ by utilizing the Feature Extractor Module, Multi-Scale Feature Fusion Module and Hybrid Co-Attention Module with Equations (1)–(4);3:Obtain feature vectors ${\{z}_{1}^{cpp},z_{2}^{cpp}\}$$\;\mathrm{and}\;{\{z}_{1}^{csp},z_{2}^{csp}\}$ by utilizing the Complementarity Projection Head and Consistency Projection Head with Equations (5) and (6);4:Concatenate $\left\{f_{1}^{gap},f_{2}^{gap}\right\}$ and input into the Classier Module to obtain final outputs p with Equation (7);5:Calculate $L_{focal}$ with Equations (9)–(11) and $L_{comple},\;L_{consist}$ with Equations (12)–(14); then, calculate L with Equation (8);6:Update the parameters $\theta_{FEM},\theta_{MFFM},\theta_{HCAM},\;\theta_{CPP},\theta_{CSP},\theta_{CLS}$ of the model by utilizing the Gradient Descent Algorithm;**end****Output:** trained parameters $\theta_{FEM},\theta_{MFFM},\theta_{HCAM},\;\theta_{CPP},\theta_{CSP},\theta_{CLS}$.

#### 2.3.1. Feature Extractor Module

The feature extraction module consists of two feature extractors in parallel, which are used to extract the features of three front view images and two side view images, respectively. Both feature extractors have the same structure, as shown in Figure 4. Based on the ResNet18 backbone network [21], we removed four convolutional modules and output feature maps from four different scales. The feature extraction module denoted as FEM(·) aimed to extract image features from front view groups and side view groups, respectively, as shown in Equation (1).
(1)$$f_{1}={FEM}_{1}\left(concate(x_{1})\right),\;f_{2}={FEM}_{1}(concate(x_{2}))$$
where $f_{1}=\{f_{1}^{\left(1\right)},f_{1}^{\left(2\right)},f_{1}^{\left(3\right)},f_{1}^{\left(4\right)}\}$, $f_{2}=\{f_{2}^{\left(1\right)},f_{2}^{\left(2\right)},f_{2}^{\left(3\right)},f_{2}^{\left(4\right)}\}$, $f_{1,2}^{\left(i\right)}$ represents the $i^{th}$ feature map of the multi-view image after feature extraction and indicates the operation of stacking multiple views in the channel dimension. The feature extractor outputs four multi-scale feature maps in four different network depths. The feature map output from the shallow layer of the network has a small receptive field and mainly contains the local information of the image, while the feature map output from the deep layer of the network has a larger receptive field and mainly contains global information of the image. By extracting multi-scale image features, the model can understand the image content more comprehensively, enhance the semantic information of the image, and improve the generalization ability and robustness of the model.

#### 2.3.2. Multi-Scale Feature Fusion Module

The fusion of the four multi-scale features output from the feature extractor helps to synthesize the feature information at different scales, which improves the feature representation and enhances the performance of the model. Therefore, we propose a Multi-Scale Feature Fusion Module. By using 1 × 1 convolution to change the number of channels for shallow feature maps and down-sampling and using 1 × 1 convolution to change the number of channels for deep feature maps and up-sampling, four feature maps with the same dimension are obtained. Then, these four feature maps are stacked in the channel dimensions for feature fusion. The Multi-Scale Feature Extraction module is denoted by MFFM(·), and four multi-scale feature maps are fused by ${MFFM}_{1}(\cdot )$ and ${MFFM}_{2}(\cdot )$ to obtain feature maps ${\{f}_{1}^{fusion},\;f_{2}^{fusion}\}$, as shown in Equation (2). Figure 5 shows the structure of the Multi-Scale Feature Fusion Module, illustrating the specific steps and realization of the fusion process. Through this feature fusion method, the model can make full use of the feature information in different scales of the image, which improves the model’s ability to express the features of the image and effectively improves the performance of the model.

#### 2.3.3. Hybrid Co-Attention Module

The Hybrid Co-Attention Module consists of the Channel Co-Attention Module and the Spatial Co-Attention Module, as shown in Figure 6. The Channel Co-Attention Module performs pooling operations on the fused feature maps using pooling kernels of sizes (H,1) and (1,W), respectively, and resizes the feature maps to $f_{H}^{1,2}\in R^{H\times C}$ and $F_{w}^{1,2}\in R^{W\times C}$. Then, we can use one-dimensional convolution and the Sigmoid activation function to obtain channel weights corresponding to horizontal and vertical features. The input feature map is weighted with the channel weight and stacked in the channel dimension to obtain the output feature map $f_{C}^{1,2}\in R^{2C\times H\times W}$. The Channel Co-Attention Module is designed to highlight feature channels that are important to the current task and suppress task-irrelevant feature channels by weighting each channel of the feature map in both vertical and horizontal directions, respectively. Such a weighting operation can make the network focus more on the feature channels with higher weights, thus improving the performance of the model. Since the input of the Hybrid Co-Attention Module is the feature map obtained by the feature extractor after the stack of different views in channel dimensions, the Channel Co-Attention Module can automatically determine the weight of each view and make full use of the importance between different views.

The Spatial Co-Attention Module uses average pooling, maximum pooling, and one-dimensional convolution operations to aggregate features for input feature graphs in the channel dimension. Then, the convolution kernels of sizes (H,1) and (1,W) are used to extract the features in the horizontal and vertical directions, respectively, and the Sigmoid activation function is used to obtain the position weights of the feature map. The input feature map is weighted according to the position weight and stacked in the channel dimension to obtain the output feature map $f_{P}^{1,2}\in R^{2C\times H\times W}$. The Spatial Co-Attention Module can make the model pay more attention to the areas of interest, namely, the mouth, neck, and other areas associated with difficult airways, while ignoring irrelevant areas such as the background. Finally, $f_{C}^{1,2}$ and $f_{P}^{1,2}$ are stacked in the channel dimension and the feature map size is adjusted to the input feature map size using one-dimensional data.

The Hybrid Co-Attention Module is denoted by *HC*AM(·), and the feature map after fusion $\left\{f_{1}^{att},f_{2}^{att}\right\}$ is obtained by the Hybrid Co-Attention Module, as shown in Equation (2).
(2)$$f_{1}^{att}={HCAM}_{1}\left(f_{1}^{fusion}\right),\;{\;f}_{2}^{att}={HCAM}_{2}(f_{2}^{fusion})$$

The Hybrid Co-Attention Module can make the model focus on important information, thus improving the model’s ability to express the input data. By dynamically assigning weights, the model can more effectively capture key features in the data, reduce the sensitivity to noise and useless information, and improve the flexibility and adaptability of the model.

#### 2.3.4. Classifier Module

The classifier module consists of two fully connected layers, two dropout layers with a dropout rate equal to 0.5, and the Sigmoid activation function. The dropout layers are between the two fully connected layers. The purpose of the classifier is to form a global representation of the input data by integrating and mapping the features extracted from the previous modules. For the two feature maps $f_{1}^{att}$ and $f_{2}^{att}$, the global average pooling is carried out, respectively, and reshaped to obtain the one-dimensional eigenvector $f_{1,2}^{gap}\in R^{C}$, as shown in Equation (3). For two eigenvectors $f_{1,2}^{gap}$, on the one hand, the complementary prediction head and the consistent prediction head, denoted by $CPP\left(\cdot \right)$ and $CPP\left(\cdot \right)$ respectively, are input to obtain the eigenvectors ${\{z}_{1}^{cpp},z_{2}^{cpp}\}$ and ${\{z}_{1}^{csp},z_{2}^{csp}\}$, as shown in Equations (4) and (5).
(3)$$f_{1}^{gap}=GAP\left(f_{1}^{att}\right),f_{1}^{gap}=GAP\left(f_{1}^{att}\right)$$
(4)$$z_{1}^{cpp}=CPP(f_{1}^{GAP}),\;z_{2}^{cpp}=CPP(f_{2}^{GAP})$$
(5)$$z_{1}^{csp}=CSP(f_{1}^{GAP}),\;z_{2}^{csp}=CSP(f_{2}^{GAP})$$
Both $CPP\left(\cdot \right)$ and $CSP\left(\cdot \right)$ are composed of a linear layer, and the obtained eigenvectors ${\{z}_{1}^{cpp},z_{2}^{cpp}\}$ and ${\{z}_{1}^{csp},z_{2}^{csp}\}$ are used to calculate consistency loss and complementarity loss, respectively. On the other hand, $f_{1,2}^{gap}$ is stacked, and the Classifier Module is input, which is denoted by CLS(·) and finally, the prediction probability p is output for each category, as shown in Equation (6).
(6)$$p=CLS(concate(f_{1}^{gap},f_{2}^{gap}))$$

### 2.4. Objective Loss Function

The objective loss function designed for this study is defined as follows:(7)$$L=L_{focal}+{\lambda_{1}L}_{comple}+{\lambda_{2}L}_{consist}$$

Among them, $L_{focal}$ is the Focal Loss, which is proposed by Kaiming He [22]. $L_{consist}$ and $L_{comple}$ are the consistency loss and the complementarity loss, respectively, that we designed based on the characteristics of the multi-view data. $\lambda_{1}$ and $\lambda_{1}$ are hyperparameters used to balance the three types of losses, which are set to 0.25 and 0.5 in this research, respectively. The consistency of information among multi-view data means that different views in multi-view data describe the same object or entity, so there is an intrinsic consistency among these views. The complementarity of information among multi-view data means that each view contains information or features that other views do not have, and these additional information or features can complement each other to provide a more comprehensive and accurate description of the object or entity. The function of the objective loss function designed in this research is to maximize the information that is not shared between the frontal views and the side views and constrain the information shared between them to ensure the consistency of their information so that the network extracts more effective features and improves the accuracy of the model. The complementarity and consistency of information are shown in Figure 7.

#### 2.4.1. Focal Loss

Focal Loss is mainly used to solve the problem of category imbalance, which is an improvement of the cross-entropy loss function, introducing a hyperparameter on the basis of the cross-entropy function to increase the weight of samples with fewer categories, and adjusting the weight relationship between the easy-to-categorize samples and the difficult samples. By decreasing the weight of the easy-to-categorize samples, the Focal Loss allows the model to focus more on the difficult-to-categorize samples during training and adds a balancing parameter to adjust the weight of the positive and negative samples. The loss function is defined as follows:(8)$$L_{focal}=-\alpha_{t}{\left(1-p_{t}\right)}^{\gamma }\log\left(p_{t}\right)$$
where $p_{t}$ is denoted by the degree of difficulty in categorizing the sample, defined as follows:(9)$$p_{t}=\left\{\begin{array}{cc} p & if\;y=1\\ 1-p & otherwise \end{array}\right.$$
(10)$$\alpha_{t}=\left\{\begin{array}{cc} \alpha & if\;y=1\\ 1-\alpha & otherwise \end{array}\right.$$
where p is the predicted value of the model output, γ is the hyperparameter for determining the loss attenuation, and α is the hyperparameter for balancing the positive and negative sample weights, which are set to 2.0 and 0.25 in this research, respectively.

#### 2.4.2. Complementarity Loss

Complementarity loss is used to maximize information that is not shared between different views. Specifically, complementarity loss is used to maximize the information that is not shared between the front view and the side view in this research, which is defined as follows:(11)$$L_{comple}=\frac{1}{N}\sum_{i}^{N}max(0,\tau -{dis\left(z_{i,1}^{cpp},z_{i,2}^{cpp}\right)}^{2})$$
where *τ* is the hyperparameter, which is set to 1 in this research. dis(·) is the distance function, which is used to measure the similarity of two vectors. We used Euclidean distance as the distance function in this research, which is defined as follows:(12)$$dis\left(\cdot \right)={\left\|x-y\right\|}_{2}$$

#### 2.4.3. Consistency Loss

Consistency loss is used to constrain the consistency of information shared between different views to ensure the consistency of their information. Specifically, consistency loss is used to constrain the information shared between front and side views in this research, which is defined as follows:(13)$$L_{consist}=\frac{1}{N}\sum_{i}^{N}{dis\left(z_{i,1}^{csp},z_{i,2}^{csp}\right)}^{2}$$

## 3. Results

### 3.1. Experiment Settings

As mentioned above, our dataset contains a total sample of 225 patients, including 200 non-difficult airway patients and 25 difficult airway patients. For model training, we used 80% of the data (180 patients, including 160 non-difficult airway patients and 20 difficult airway patients). For model testing, we used the remaining 20% of the data (45 patients, 40 non-difficult airway patients, and 5 difficult airway patients). To ensure the validity and reliability of the model, we used a five-fold cross-validation method to reduce the dependence of the results on the specific partitioning of the dataset, as shown in Figure 8. In the five-fold cross-validation, the dataset was first divided uniformly and randomly into five non-overlapping folds, each containing about 20% of the data. These five folds were then sequentially used as a test set, while the remaining four folds were combined as a training set to train the model. In this way, we could obtain five different training–testing data partitions that had the same training–testing ratio, but each time, the training set and the testing set were different. For each partition, we trained a model independently and evaluated its performance using the testing set, ensuring that each model was validated on a completely independent testing set, increasing the reliability of the evaluation. Finally, we collected the test results of the five models on their respective testing sets and calculated the average. The average reflected the comprehensive performance of the model on different testing sets and could be used as a reliable indicator of the overall performance of the model. Through five-fold cross-validation, we were able to more accurately evaluate the generalization ability of the model and reduce the risk of overfitting and underfitting.

In the experiments, we set 120 epochs, using Adam [23] as the optimizer, and the batch size to 16. In order to adjust the learning rate more efficiently, we used a segmented constant decay strategy. Specifically, we set the initial learning rate to 0.001 and adjusted the learning rate to 0.1 times that of the original after every 30 epochs. This strategy helped to quickly approach the optimal solution with a large learning rate at the beginning of training and fine-tune the model with a small learning rate at the end of training, which prevented the model from oscillating near the optimal solution. In order to avoid the overfitting of the model during the training process, we adopted the early stopping strategy. Specifically, if the value of the loss function of the model did not decrease further in five consecutive epochs, we stopped the training and saved the current best model parameters. All experiments were performed on Windows 10. The GPU was NVIDIA GeForce RTX 4080, and the CPU was 13th Gen Intel Core i5-13600KF. Miniaconda 23.1.0 was used to manage the Python environment to ensure the consistency and repeatability of the experiments. In the Python environment, we chose Python version 3.8 and used PyTorch 2.1.0 as the deep learning framework.

### 3.2. Evaluation Metrics

In this research, we used accuracy, sensitivity, specificity, and *F*_1_ score as the overall performance metrics. These evaluation metrics are defined as follows:(14)$$Acc=\frac{TP+TN}{TP+TN+FN+FP}$$
(15)$$Spe=\frac{TN}{TN+FP}$$
(16)$$Sen=\frac{TP}{TP+FN}$$
(17)$$F_{1}=\frac{2\times TP}{2\times TP+FP+FN}$$
where TP, TN, FP, and FN denote a true positive, true negative, false positive, and false negative, respectively. In medical image classification tasks, in addition to conventional evaluation metrics, the Receiver Operating Characteristic (ROC) curve and its related Area Under The Curve (AUC) are key measures of model performance. The ROC curve provides a comprehensive and intuitive view to evaluate the performance of binary classifiers. It demonstrates the trade-off between the true case rate and the false positive case rate at different classification thresholds. Specifically, the ROC curve is plotted by continuously tracking all possible sensitivity–specificity pairs, and each point on the ROC curve corresponds to a specific classification threshold, reflecting the classification performance of the model under this threshold. It means that no matter how we chose the classification threshold, the ROC curve gave the overall performance of the model rather than being limited to a particular threshold setting. The AUC value is a quantitative metric used to evaluate the overall performance of the model. The AUC value usually ranges from 0.5 to 1, and the closer the AUC value is to 1, the better the model is at distinguishing between positive and negative samples, i.e., the better the model’s performance.

### 3.3. Experiment Analysis

Based on the five-fold cross-validation method, we independently trained five classification models and tested them separately. Figure 9 shows the ROC curves of each model, and the corresponding AUC was calculated. The ROC curves of each validation in the figure showed that the model could still maintain a high true positive rate under a low false positive rate, indicating that its performance is better, and the AUC values obtained from the five cross-validations are relatively stable, which are all in the range of 0.73 to 0.82. It indicates that the performance of the model has a certain degree of stability and reliability. The subsequent experimental results are expressed by the average of the five cross-validation results.

To validate the effectiveness of our proposed method, we reproduced four state-of-the-art (SOTA) deep learning-based difficult airway assessment methods and trained and tested them according to the same experimental setup as our experiments. The performance of these four methods was compared with the performance of our proposed method in Table 1. Our proposed method provides a significant improvement over the current deep learning-based SOTA method for difficult airway assessments. Specifically, our method achieved the highest performance in all metrics, with an absolute advantage of 10.00% and 7.96% achieved in Sen and *F*_1_, respectively, demonstrating the excellent performance of our method for difficult airway assessments and proving its effectiveness in airway assessment tasks.

We investigated the classification performance of different types of backbone networks, including AlexNet [24], GoogLeNet [25], DenseNet121 [26], MobileNetV_2 [27], ShuffleNetV_2 [28], VGG16 [29], ResNet18 and the backbone networks we used. For different types of backbone networks, all experimental settings were kept the same. As shown in Table 2, we can observe that the best experimental results were obtained using the ResNet 18 and DenseNet121 backbone network. However, the performance of our proposed backbone was very close to that of ResNet18 and DenseNet121, and the number of parameters of our backbone was 43% and 70% of that of ResNet18 and DenseNet121, respectively, and FOLOPs were 49% and 31% that of ResNet18 and DenseNet121, respectively. It shows that the model we are using achieved optimal lightweight levels while maintaining performance, which can meet the needs of future mobile deployments.

We then investigated the effect of using different attention mechanisms on the network classification performance, including SE Attention [30], CBAM Attention [31], ECA Attention [32], and CA Attention [33] and the Hybrid Co-Attention Module, which we used in this research. For the different attention mechanisms, all experimental settings were kept the same. From Table 3, we can observe that although the sensitivity index of the Hybrid Co-Attention Module was 1.25% lower than that of ECA, the remaining indexes reached the highest. Without the use of the attention mechanism, accuracy, specificity, sensitivity, and *F*_1_ score were improved by 4.95%, 3.12%, 7.5%, and 6.13%, respectively. The reason for this is that the Hybrid Co-Attention Module combines channel attention and spatial attention, which makes the model pay attention to the important semantic information of the image in both the vertical direction and horizontal direction, improving the performance of the model.

In addition, to verify the effectiveness of our multi-view image fusion method, we compared the performance of the model using multi-view images and single-view images as input. As shown in Table 4, the fusion of 5 images produced optimal results, with all performance metrics higher than the corresponding performance metrics of the single-view image model. It demonstrates the feasibility of a multi-channel information fusion method. The method takes into account information from different views of the patient, enabling more comprehensive information to be captured with higher reliability.

### 3.4. Ablation Study

In order to validate the effectiveness of each module in the model proposed in this study and to deeply understand the impact of different modules of the model on the overall performance, we conducted a series of ablation experiments. By gradually removing specific modules or loss functions of the model, including the Multi-Scale Feature Fusion Module, Hybrid Co-Attention Module, Focal Loss, complementarity loss, and consistency loss, we observed and analyzed the effects of these changes on the model performance. We used the base multi-view classification model as the baseline, and the feature extraction module of the baseline model was the same as that of the model proposed in this study, where the multi-view features were extracted at the last scale and then fused, two fully connected layers were added for classification, and binary cross-entropy loss was used. Table 5 shows the quantitative results of the ablation experiments. After removing the consistency loss and complementarity loss, the *F*_1_-score of the model decreased by 3.97%, and Sen decreased by 3.75%, which illustrates the necessity of our work and how the introduction of consistency loss and complementarity loss can make full use of the information of multi-view data and improve the accuracy of classification. After removing HCAM, the *F*_1_-score of the model decreased by 1.44%, and Sen decreased by 1.50%, which indicates that the Hybrid Co-Attention Module adopted in this study can make the model focus on the region of interest and further improve the performance of the model. Without MFFM, the *F*_1_ of the model decreased by 2.82%, and Sen decreased by 2.25%, which confirms that the Multi-Scale Feature Fusion Module enables the model to fully take into account both coarse-grained and fine-grained features of the image, which significantly improves the performance of the model.

## 4. Discussion

It has been shown that patients with difficult airways exhibit subtle differences in facial features from those with non-difficult airways. Mouth opening is defined as the distance between the upper and lower mesial incisors at the maximum mouth opening. Patients with a mouth opening of less than 3 cm have a higher risk of developing a difficult airway. The head and neck range of motion is defined as the degree of extension of the atlantoaxial joint, and joint extension is associated with a reduced difficult airway. The visibility of intraoral structures in the presence of an open mouth and extended tongue is related to a difficult airway and can be categorized into four levels of visibility by the Mallampati method: Level I: the patient’s tonsils, palatal sails, and soft palate can be clearly visualized. Grade II: the patient’s tonsils and part of the palatal sail can be seen, as well as part of the uvula. Grade III: only the root of the soft palate and uvula can be seen. Grade IV: Only the hard palate can be visualized. The higher the Mallampati grade, the more likely the patient is to have a difficult airway. The nail–chin distance is defined as the distance from the incisura of the thyroid cartilage to the tip of the mandible when the neck is fully extended, with normal values for adults being in the range of 6.5 cm or more. Less than 6 cm or the width of the examiner’s third horizontal finger suggests that the patient may have a difficult airway. In facial proportions, the distance from brow to nose tip and distance from the apex of the coronal plane of the frontal squamosal body surface to the chin is related to a difficult airway, and the greater the proportion, the greater the possibility of a difficult airway [34]. In summary, the differences in facial features between patients with difficult airways and those without difficult airways are mainly in the mouth and neck regions.

Therefore, in this research, we propose a Dual-Path Multi-View Fusion Network. It aims to combine the multi-view learning method to capture the different features of the mouth and neck through the multi-view facial images of the patient and predict whether the patient has difficult airways. Concretely, in terms of model structure, the network adopts a dual-branch structure, which can be used to extract features from the front view and the side view, respectively, so that each branch can focus on extracting key features from its own view, thus improving the efficiency and accuracy of feature extraction. In addition, the model uses Hybrid Co-Attention. By dynamically assigning weights, the model can capture key features in the data more effectively, reduce the sensitivity to noise and useless information, and improve the flexibility and adaptability of the model. In terms of loss function, consistency loss and complementarity loss are designed to measure the consistency and complementarity of information between different views of learning so as to improve the accuracy of difficult airway assessments. In order to deal with the imbalance of positive and negative samples, a series of strategies are adopted, including using Focal Loss, the data enhancement of positive and negative samples in different proportions, and adding dropout layers. In the ablation experiment, the effectiveness of each module is demonstrated by the test results by gradually removing each module. This novel deep learning-based method for difficult airway assessment demonstrates excellent performance comparable to expert judgment.

## 5. Conclusions

In this research, we propose a Dual-Path Multi-View Fusion Network for difficult airway assessments by extracting multi-view facial image features from patients. It provides a reliable auxiliary tool for clinical staff to effectively improve the accuracy and reliability of preoperative difficult airway assessments and helps them to recognize and assess the risk of difficult airways before operation so as to take appropriate preventive measures and reduce the incidence of postoperative complications. According to the characteristics of the dataset, we designed a dual-path model structure by combining the idea of multi-view learning, grouping frontal and lateral images to extract the features, adding the Multi-Scale Feature Fusion Module and Hybrid Co-Attention Module to improve the feature expression ability of the model, using consistency loss and complementarity loss, which make full use of the complementarity and consistency of information between the multi-view data, and then combining Focal Loss. The three losses constrain each other and effectively prevent information bias. This research not only provides valuable insights to address the challenges associated with multi-view learning in medical image analysis but also provides a powerful framework for future research in this area. However, there are still some limitations in this research. This model suffers from potential sample bias due to the presence of irregular criteria for physicians to capture patient facial images, the complex background noise of the images, and a small amount of data, and the performance of the model is largely limited by the quality and quantity of the dataset. In future research, we will further standardize the criteria for data collection, expand the dataset, especially to include a more diverse sample set of airway difficulties, and incorporate multi-modal data such as patients’ height, weight, and medical history to further enhance the generalization and robustness of the model.

In future studies, we will collect patients’ multi-modal information, including patients’ personal physical indicators, medical records, and other text data, patients’ audio data, and patients’ multi-view image data, and use multi-modal fusion to further improve the accuracy and reliability of difficult airway assessments. Based on the demand analysis of hospitals, we plan to develop a difficult airway management system that is deployed on a mobile terminal system.

## Figures and Tables

**Figure 1 bioengineering-11-00703-f001:**
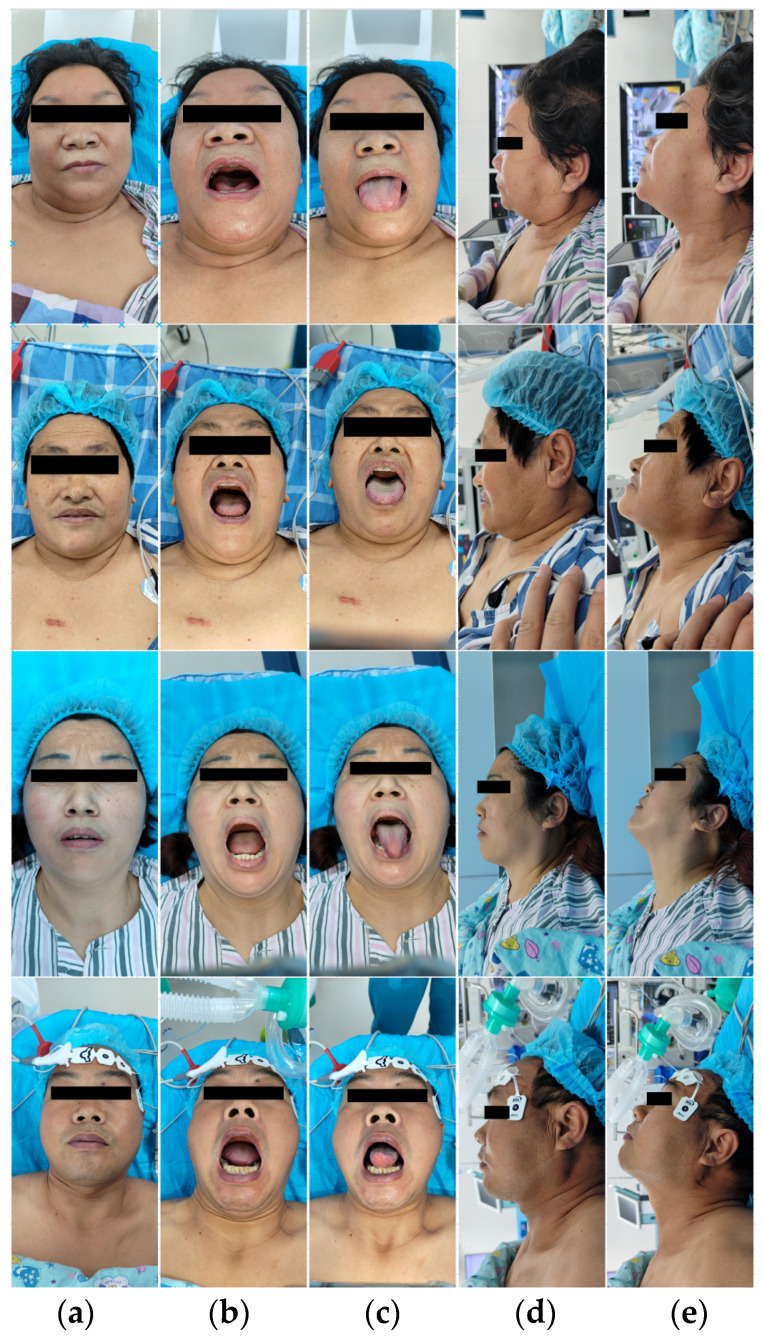
Partial patient images in the dataset. Each patient is shown from five different views: (**a**) frontal closed-mouth base position; (**b**) frontal open-mouth base position; (**c**) frontal tongue-protruding base position; (**d**) lateral closed-mouth base position; and (**e**) lateral closed-mouth chin-lift position. The periocular area of the patients was mosaiced to protect the privacy of the patients.

**Figure 2 bioengineering-11-00703-f002:**
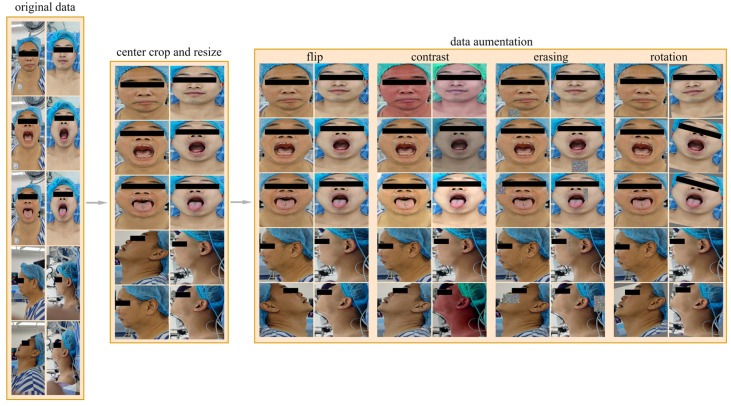
Data preprocessing process. The data preprocessing process includes center cropping, resizing, and data augmenting, and the data augmentation methods we used included random horizontal flipping, random changes in brightness, contrast, and saturation, random rotations within the range of [−15°, 15°], and random erasing.

**Figure 3 bioengineering-11-00703-f003:**
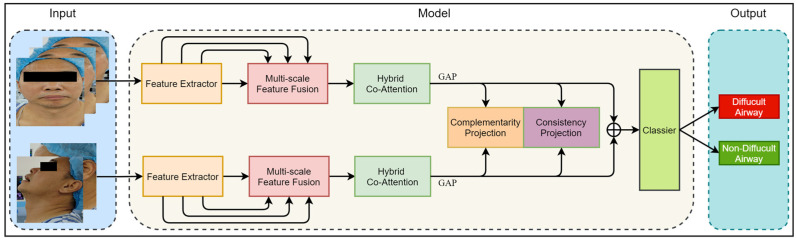
The overall framework of the model. The input of the model includes five views of the image divided into two groups: the front view and the side view. The model consists of a Feature Extractor Module, Multi-Scale Feature Fusion Module, Hybrid Co-Attention Module, and Classier Module. The output of the model is the category of difficult airway or non_difficult airway.

**Figure 4 bioengineering-11-00703-f004:**
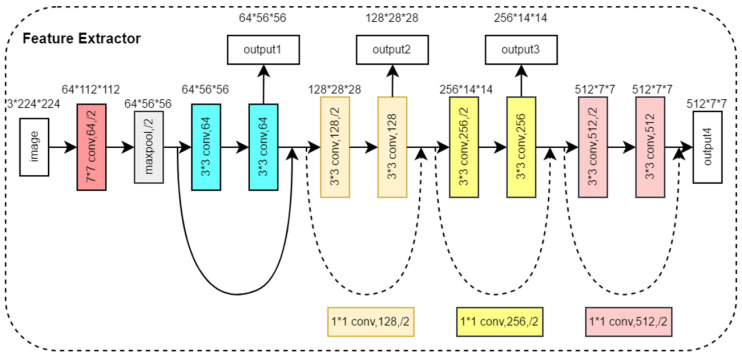
Structure of the Feature Extractor Module.

**Figure 5 bioengineering-11-00703-f005:**
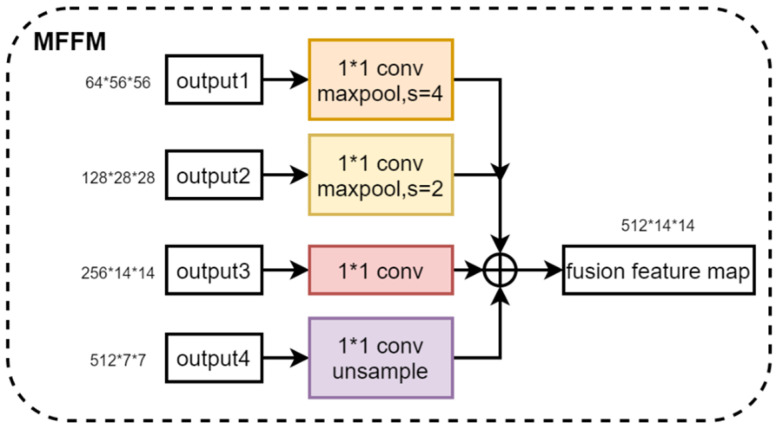
Structure of the Multi-Scale Feature Fusion Module.

**Figure 6 bioengineering-11-00703-f006:**
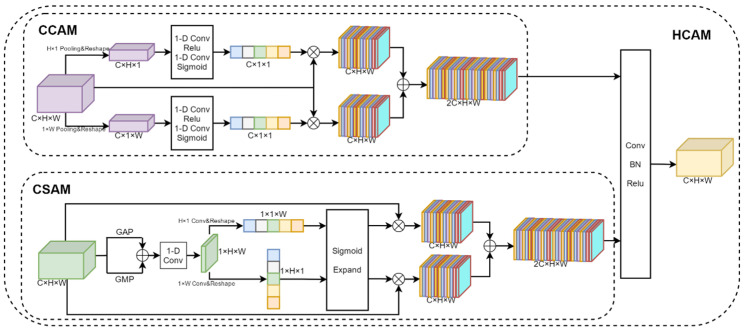
Structure of the Hybrid Co-Attention Module. It consists of the Channel Co-Attention Module and the Spatial Co-Attention Module.

**Figure 7 bioengineering-11-00703-f007:**
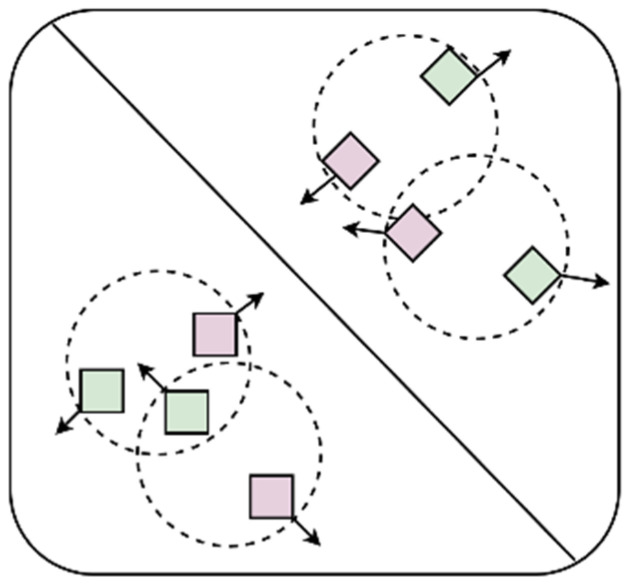
Consistency and complementarity of information. Different shapes represent different categories, and different colors represent different views. The complementarity of information means that different views of the same object can provide different information, so they should be as far away as possible in the feature space, which is represented by arrows that are far away from each other. The consistency of information means that different views of the same object are descriptions of the same object and have consistency, so they should be constrained within a certain range in the feature space, which is represented by circular dotted lines in the figure.

**Figure 8 bioengineering-11-00703-f008:**
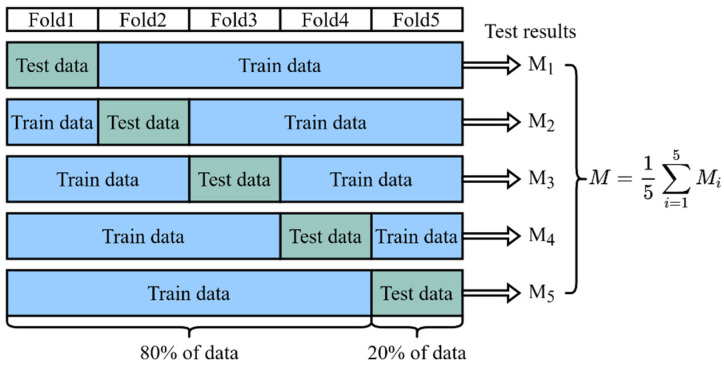
Five-fold cross-validation. The result of the final validation is obtained from the average of the five cross-validations.

**Figure 9 bioengineering-11-00703-f009:**
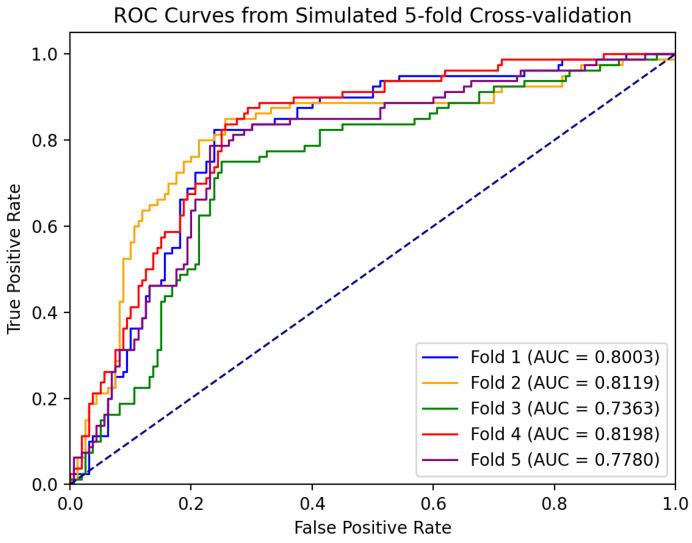
ROC curve of five-fold cross-validation. The results of the five cross-validations are represented by five different colored lines.

**Table 1 bioengineering-11-00703-t001:** Comparison of state-of-the-art methods for difficult airway assessments.

Methods	Acc/%	Spe/%	Sen/%	*F*_1_/%
Wang et al. [18]	70.42	71.25	68.75	60.77
Hayasaka et al. [16]	65.83	69.37	58.75	53.40
Tavolara et al. [17]	65.42	65.62	65.00	55.61
García-García et al. [19]	72.08	71.88	72.50	63.39
Ours	**77.92**	**75.62**	**82.50**	**71.35**

**Table 2 bioengineering-11-00703-t002:** Comparison of metrics for different backbones.

Backbone	Acc/%	Spe/%	Sen/%	*F*_1_/%	Params/M	FLOPs/G
AlexNet	69.17	69.38	68.75	59.78	2.47	0.66
GoogleNet	72.50	71.88	73.75	64.13	5.60	1.51
DenseNet121	78.33	75.62	**83.75**	**72.04**	6.95	2.90
MobileNet_v2	71.67	71.25	72.50	63.04	2.22	0.33
ShuffleNet_v2	72.92	71.89	75.00	64.86	**1.25**	**0.15**
VGG16	75.42	73.75	78.75	68.10	14.17	15.34
ResNet18	**78.33**	**76.87**	81.25	71.43	11.18	1.82
Ours	77.92	75.62	82.50	71.35	4.91	0.90

**Table 3 bioengineering-11-00703-t003:** Comparison of metrics for different attention mechanisms.

Attention Mechanism	Acc/%	Spe/%	Sen/%	*F*_1_/%
—	73.33	72.50	75.00	65.22
SE	74.17	73.12	76.25	66.30
CBAM	75.00	74.38	76.25	67.03
CA	74.17	72.50	77.50	66.67
ECA	76.67	73.12	**83.75**	69.52
HCAM (Ours)	**77.92**	**75.62**	82.50	**71.35**

**Table 4 bioengineering-11-00703-t004:** Comparison of metrics for different input cases: multi-view images and single-view images as input.

	Acc/%	Spe/%	Sen/%	*F*_1_/%
View 1	60.00	57.50	65.00	52.00
View 2	65.83	68.75	60.00	53.93
View 3	62.50	60.00	67.50	54.55
View 4	57.50	45.00	82.50	56.41
View 5	65.00	61.25	72.50	58.00
View 1~5	**77.92**	**75.62**	**82.50**	**71.35**

**Table 5 bioengineering-11-00703-t005:** The assessment of performances of different modules in our framework. Before adding our proposed modules, we used the base multi-view classification model as the baseline. The feature extraction module of the baseline model is the same as that of the model proposed in this study, where the multi-view features are extracted at the last scale and then fused; two fully connected layers are also added for classification.

Component	Choice						
Baseline	✓	✓	✓	✓	✓	✓	✓
MFFM		✓	✓	✓	✓	✓	✓
HCAM			✓	✓	✓	✓	✓
Focal Loss				✓	✓	✓	✓
Complement loss					✓		✓
Consistence loss						✓	✓
Acc/%	69.17	71.66	73.33	74.58	75.42	75.42	**77.92**
Spe/%	68.75	71.25	73.12	72.50	73.12	73.75	**75.62**
Sen/%	70.00	72.25	73.75	78.75	80.00	78.75	**82.50**
*F*_1_/%	60.22	63.04	64.48	67.38	68.45	68.11	**71.35**

## Data Availability

The research data is unavailable due to ethical restrictions.

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
