# Peer review of "Difficult Airway Assessment Based on Multi-View Metric Learning"

_bioengineering, 2024, doi:10.3390/bioengineering11070703_

Round 1
Reviewer 1 Report
Comments and Suggestions for Authors
Thank you for sharing such interesting study. Here are my comments below.
1. Table 3 shows that DenseNet121 is better than or the equal to the performance of the proposed. It is interesting to see the performance of attention based NET such as attention-56. Since the results are very similar. It should be provided the number of parameters used in the backbone in order to show the efficiency of the proposed NET.
2. Although many strategies implemented prevent overfitting and bias, overfitting strategy likes dropout rate should be applied. The proposed Net should be further tuned with dropout rate in the backbone’s last few layers in order to further improve and handle overfitting issue.
3. The discussion section is too short. The novelty should be lighted at the beginning of the section. Good discussion of NET key features and the ablation results should be provided.
4. Vision-Transformer class NET has not included in this study. It should be discussed the reason.
5. Future plan is too simple. It should also discuss about the potential next version of the proposed design.
Author Response
Comments 1: Table 3 shows that DenseNet121 is better than or the equal to the performance of the proposed. It is interesting to see the performance of attention based model such as attention-56. Since the results are very similar. It should be provided the number of parameters used in the backbone in order to show the efficiency of the proposed model.
Response 1: Thank you very much for the feedback. We have provided the number of parameters used in the backbone in order to show the efficiency of the proposed model. Specifically, we added a new column in Table 3 to show the number of parameters for each backbone, and recalculated corresponding FLOPs. Meanwhile, we demonstrate the efficiency of our model by comparing densenet121 with the number of parameters and FLOPs of our model. This change can be found at Page 14, Paragraph 2 and Table 3.
Comments 2: Although many strategies implemented prevent overfitting and bias, overfitting strategy likes dropout rate should be applied. The proposed Net should be further tuned with dropout rate in the backbone’s last few layers in order to further improve and handle overfitting issue.
Response 2: Thank you very much for your suggestion. In fact, we used dropout in the proposed model. Specifically, we set two dropout layers with dropout rate equal to 0.5 in the two fully connected layers of the classifier module. However, we are sorry that we ignored the description in the paper. We have added description of the dropout layer at the corresponding location in the paper. This change can be found at Page 9, Paragraph 3.
Comments 3: The discussion section is too short. The novelty should be lighted at the beginning of the section. Good discussion of NET key features and the ablation results should be provided.
Response 3: Thank you very much for your suggestion. We have expanded the discussion section appropriately. Specifically, we have introduced the innovations of the proposed model at the latter part of the section and added a description of the key features of the proposed model and the results of the ablation experiment. This change can be found at Page 16, Paragraph 3.
Comments 4: Vision-Transformer class NET has not included in this study. It should be discussed the reason.
Response 4: Thank you very much for your suggestion. Vision-Transformer is a well-known and excellent model in recent years, but it is not suitable for our task. Because the effect of Vision-Transformer largely depends on the huge amount of data, the number of dataset in our task is very limited. Therefore, the Vision-Transformer class NET has not included in this study.
Comments 5: Future plan is too simple. It should also discuss about the potential next version of the proposed design.
Response 5: Thank you very much for your suggestion. We have discussed our future plans in more detail at the end of the section 5. This change can be found at Page 17, Paragraph 3.
Reviewer 2 Report
Comments and Suggestions for Authors
27.06.2024
A review to evaluate its suitability for publication Type of manuscript:
Article
Title: Difficult Airway Assessment Based on Multi-view Metric
Learning
Authors: Jinze Wu , Yuan Yao , Guangchao Zhang , Xiaofan Li , Bo Peng *
The Paper presented by the Bo Peng and co-authors in Biosignal Processing Section of Bioengineering MDPI, focused on the method Difficult Airway Assessment Based on Multi-view Metric Learning.
The network proposed by the authors may help in identifying and assessing the risk of difficult airway patency in patients before surgery, which is the current topic of this study.
The manuscript combines medical aspects and mathematical and chemometric processing of the results using the Dual-path Multi-view Fusion Net work model. For this purpose, a special algorithm of actions was developed taking into account the test set. Five-fold cross-validation test allowed to evaluate the proper quality of the model work.
The manuscript gives the impression of a properly prepared paper with a detailed description of the developed chemometric approaches and estimation of statistical parameters.
There are the comments:
1. How the model will behave when new data are added,?
2. A note on layout: the Materials and Methods section appears twice - number 2 and number 3. This should be removed by combining the two sections.
3. Given the presentation of photographs of real patients, do the authors have ethical approval for such photographs?
4. Design Note: The Materials and Methods section appears twice - number 2 and number 3. This should be removed by combining the two sections.
5. Given the presentation of photographs with real patients, is there ethical approval for such photographs?
Respectfully, reviewer
Author Response
Comments 1: How the model will behave when new data are added?
Response 1: Thank you for your question. Because the number of dataset used in our study is very limited, when new data is added, the number of positive and negative samples contained in the data set will increase correspondingly, which means that the model can learn more features and patterns from it, reducing the risk of model overfitting, and helping improve the generalization ability of the model. We will try to collect more data to enhance the model performance in the near future.
Comments 2: A note on layout: the Materials and Methods section appears twice - number 2 and number 3. This should be removed by combining the two sections.
Response 2: Thank you for pointing this out. We made a writing mistake, the title of the section 3 in the paper should be "Results", we mistakenly wrote the name of the second 2 "Materials and Methods", we have changed the title of the section 3 to "Results". This change can be found at Page 11.
Comments 3: Given the presentation of photographs of real patients, do the authors have ethical approval for such photographs?
Response 3: Thank you very much for your attention to the issue of showing photos of real patients in my paper. First, we fully understand the importance of ethical approval when displaying photographs of real patients. When collecting the patient's image, we have clearly informed the patient that the photo will be used for the purpose of scientific research and academic exchange, and have obtained the patient's consent. When presenting images in the paper, we put a black mask around the eyes of each patient's picture to protect the patient's privacy. Secondly, we always follow the principle of privacy in handling these photos. The photos are securely stored on a protected server, preventing unauthorized access and disclosure. The use of the photos is strictly limited to prior notice and consent, and will not be used for any commercial purposes or other unauthorized activities. As researchers, we always uphold academic integrity and ethical responsibility, and we are committed to continue to follow relevant ethical norms and laws and regulations to ensure that the rights and interests of patients are fully protected.
Comments 4: Design Note: The Materials and Methods section appears twice - number 2 and number 3. This should be removed by combining the two sections.
Response 4: Thank you for pointing this out. We made a writing mistake, the title of the section 3 in the paper should be "Results", we mistakenly wrote the name of the second 2 "Materials and Methods", we have changed the title of the section 3 to "Results". This change can be found at Page 11.
Comments 5: Given the presentation of photographs with real patients, is there ethical approval for such photographs?
Response 5: Thank you very much for your attention to the issue of showing photos of real patients in my paper. First, we fully understand the importance of ethical approval when displaying photographs of real patients. When collecting the patient's image, we have clearly informed the patient that the photo will be used for the purpose of scientific research and academic exchange, and have obtained the patient's consent. When presenting images in the paper, we put a Mosaic around the eyes of the patient's picture to protect the patient's privacy. Secondly, we always follow the principle of privacy in handling these photos. The photos are securely stored on a protected server, preventing unauthorized access and disclosure. The use of the photos is strictly limited to prior notice and consent, and will not be used for any commercial purposes or other unauthorized activities. As researchers, we always uphold academic integrity and ethical responsibility, and we are committed to continue to follow relevant ethical norms and laws and regulations to ensure that the rights and interests of patients are fully protected.
Reviewer 3 Report
Comments and Suggestions for Authors
The paper is of excellent quality for either the used approach or the obtained results. I think that the manuscript deserves publication.
I have just two remarks:
Introduction is quite clear, but I suggest the authors, if possible, to strengthen the originality of their proposal.
I kindly ask the authors to revise the numbering of sections and subsections, that are not always precise. For instance, see page 7. We have subsection 3.1 and then subsections 2.3.1, 2.3.2, and so on.
At the actual state, my final opinion is “minor revisions”.
Author Response
Comments 1: Introduction is quite clear, but I suggest the authors, if possible, to strengthen the originality of their proposal.
Response 1: Thank you very much for your suggestion. We further describe our approach and illustrate its originality. Specifically, different from existing methods, the proposed method is designed in terms of model structure and loss function to make full use of the consistency and complementarity of information between multi-view data and improve the accuracy and reliability of difficult airway assessment. We have revised the latter part of the introduction. This change can be found at Page 3, Paragraph 2.
Comments 2: I kindly ask the authors to revise the numbering of sections and subsections, that are not always precise. For instance, see page 7. We have subsection 3.1 and then subsections 2.3.1, 2.3.2, and so on.
Response 2: Thank you for pointing this out. We have double-checked and revised the numbering of sections in the paper. This change can be found at Page 7.